# Preparation and *in vivo* effectiveness evaluation of heparin-loaded PLGA@PCL core-shell fiber small-diameter vascular grafts

Yonghao Xiao[1,2☯], Han Wang[2,3☯], Yuhao Jiao[4,5], Yuehao Xing[4,6], Lin Ye[2], Ai-ying Zhang[2], Xue Geng[2], Fanshan Qiu[3], Zengguo Feng[2]*, Hongbo Chen[3]*, Yongquan Gu[4]*

**1** College of Light Industry Science and Engineering, Tianjin University of Science and Technology, Tianjin, China, **2** School of Materials Science and Engineering, Beijing Institute of Technology, Beijing, China, **3** National Institute for Food and Drug Contro University, Beijing, China, **4** Department of Vascular Surgery, Xuanwu Hospital, Capital Medical University, Beijin, China, **5** Intervention and Hemangioma Department, Children's Hospital Capital Institute of Pediatrics, Beijing, China, **6** Department of Cardiovascular Surgery, Beijing Children's Hospital, Capital Medical University, National Center for Children's Health, Beijing, China

☯ These authors contributed equally to this work.
* sainfeng@bit.edu.cn (ZF); chenhb@nifdc.org.cn (HC); gu15901598209@aliyun.com (YG)

## Abstract

Cardiovascular disease has become the leading cause of death. It is the common goal for researchers worldwide to develop small-diameter vascular grafts (SDVGs) which could meet clinical needs. In this study, PLGA@PCL core-shell structural fibrous SDVGs was fabricated by coaxial electrospinning process, and then the surface heparinization of the vascular material was carried out after H2N-PEG-NH2 fixed on sodium hydroxide-treated electro-spun PCL tubes. Finally, the long-term patency and tissue regeneration of the grafts were evaluated in vivo through the rabbit carotid artery replacement model. The results indicate that the heparin-modified PLGA@PCL core-shell structural fibrous SDVGs achieved long-term patency and the arrangement of collagen and elastin in the neointima was similar to the native vessel in the rabbits after 9 months. After 3 months postoperatively, endothelialization was almost complete, and vascular calcification was also observed. It can be concluded that surface heparinization is a feasible modification method for in situ tissue-engineered vascular grafts, and controlling the occurrence of vascular calcification is another important issue to be solved in the development of SDVGs, and it is also the focus of our next research work.

## Introduction

Cardiovascular disease stands as the principal cause of death globally, linked with a high rate of morbidity and mortality [1,2]. These diseases occur mainly due to the stenosis or occlusion of blood vessels resulting in reduced blood flow and insufficient

**Data availability statement:** All relevant data are within the paper.

**Funding:** This work was supported by the National Key R&D Program of China (2022YFC2409802), received by Dr. Han Wang.

**Competing interests:** NO authors have competing interests.

tissue nutrient supply, which further leads to ischemic tissue damage [3,4] Coronary artery disease (CAD) is the key component of cardiovascular disease [5]. Currently, the treatments of CAD include percutaneous coronary angioplasty, drug thrombolysis, stent intervention and vascular replacement, etc. When long-segment defects happened or the defects occured in vital organs like heart, vascular transplantation is the preferred means to restore blood supply [6]. Obtaining autologous blood vessels, as we all know, is a challenging thing and could become even harder if the patient's condition is poor.

Vascular grafts are artificial substitutes and also known as artificial blood vessels. These vascular grafts play a crucial role in clinical treatments by serving as prostheses that repair or replace dysfunctional blood vessels or reroute blood flow [7]. Vascular grafts can be categorized into two types: large-diameter vascular grafts with diameters exceeding 6 mm and small-diameter vascular grafts (SDVGs) with diameters below 6 mm [8]. At present, large-diameter vascular grafts have gained widespread use in clinical practice. The demand for SDVGs continues to rise these years [9]. However, there is only one kind of SDVGs has been approved for clinical application in the world until now [10]. The main reasons are the high rate of vascular stenosis, thrombosis, infection and other adverse reactions occurred after implantation of vascular grafts, which leads to a low long-term patency rate. The fabrication of ideal SDVGs is still a difficult subject in the world [11,12].

Electrospinning, as a low-cost method for fabricating nanofiber scaffolds, has many advantages, including easy operation and easy control of the orientation, structure and morphology of the scaffolds prepared. Electrospinning technology is currently the most used method for the preparation of artificial blood vessels, accounting for about 49% [13]. It is also a common method for preparing SDVGs. In the process of fabricating SDVGs, natural polymers, synthetic polymers and decellularized matrices are the most used materials. Among them, synthetic polymers such as Polyglycolic Acid (PGA), Polylactic Acid (PLA), poly(lactide-coglycolide) (PLGA) and Poly($\varepsilon$-caprolactone) (PCL) are the most used materials for the fabrication of small diameter vascular grafts [14]. However, grafts made by these polymers can easily lead the absorption of blood cells and plasma proteins, and then cause coagulation reaction. Therefore, how to improve the hemocompatibility of the blood contact surface and reduce the risk of thrombosis has always been the focus in the investigation of SDVGs [15].

Currently, numerous methods have been used to improve the hemocompatibility of SDVGs. These methods include physical or chemical treatment of materials, incorporation of antithrombotic molecules, biofunctionalization of surfaces to mimic endothelial properties or promote endothelialization and so on [11]. Among them, the incorporation of anti-coagulative molecules are the most frequently chosen methods. Heparin, the most famous anti-coagulative drug, can be coated or immobilized on vascular grafts [16,17]. The anticoagulant effect of it is mainly achieved through the activation of antithrombin (AT-III). Fang et al. [18] fabricated a kind of bi-layered SDVGs with an inner layer of PCL and an outer layer of TPU, and the surface was chemically modified with heparin via the 1-Ethyl-3-(3-dimethylaminopropyl)

carbodiimide (EDCI)/N- hydroxy-succinimide (NHS) chemistry. These grafts showed a good compatibility of anti-thrombosis after transplantation in rabbit carotid artery. Moreover, Zhu et al. [19] constructed a highly interconnected porous silk fibroin SDVGs by freeze-drying the mixed solution of silk fibroin and heparin. The heparin incorporated in the grafts could release in a sustained manner for about 7 days and these SDVGs showed good hemocompatibility.

In this study, PLGA@PCL core-shell structural fibrous SDVG (CS1508 SDVG) was prepared by coaxial electrospinning, and then the surface heparinization of CS1508 SDVG was carried out based on our previous work, named HCS1508 SGVD [20]. Finally, we evaluated the long-term patency and tissue regeneration of HCS1508 in vivo through the rabbit carotid artery replacement model and investigated whether it could be a viable solution for SDVGs in clinical application.

## Materials and methods

### Materials

PLGA (50:50) (Jinan Daigang Biomaterial Co.,Ltd, China), PCL (Mn = 80000 g/mol) was purchased from Sigma (St. Louis, MO, USA). Hexafluoroisopropanol (HFIP) was purchased from Aladdin (Shanghai, China). Heparin was purchased from KEH Co.,Ltd (Beijing, China), NaOH was purchased from Innochem (Beijing, China). Polyethylene glycol diamine was purchased from Sigma (St. Louis, MO, USA). N-Hydroxy succinimide ($H_2N$-PEG-$NH_2$) was purchased from Sigma (St. Louis, MO, USA).1-Ethyl-3-(3-dimethylaminopropyl) carbodiimide (EDCI) was purchased from Solarbio (Beijing, China). NHS was purchased from was purchased from Innochem (Beijing, China). 2-morpho- linoethane-sulfonic acid (MES) was purchased from Aladdin (Shanghai, China). N, N-Diisopropylethylamine (DIPEA) was purchased from Urchem (Shanghai, China). Type I collagenase was purchased from Aladdin (Shanghai, China), 1-Hydroxybenzotriazole (HOBt) was purchased from Huateng Technology Company (Beijing, China).

New Zealand rabbits were provided by Academy of Military Medical Sciences. All animals were housed in independent ventilation cages (IVC) under SPF conditions at an ambient temperature of 20–26 ℃ and humidity of 40–70%. All of the animals were allowed diet and drink water in free style. The animal experiments were approved by the Animal Experiments Ethical Committee of Fuwai Hospital.

### Fabrication of the PLGA@PCL SDVGs

The PCL solution with 15% concentration and PLGA solution with 8% were prepared using HFIP as the solvent. After stirring in a magnetic stirrer at room temperature for 6 h, ultrasonic treatment was carried out for 15 minutes. The spinning solutions were transferred into a 10 ml medical syringe. The flow rate of 21G needle in the core layer was set to 0.5 ml/h, the flow rate of 16G needle in the shell layer was set to 2.1 ml/h, the receiving distance was set to 7 cm, and the positive and negative voltage was set to 11kV and −2.5kV respectively. The rotation speed of the stainless steel metal rod (ID = 2, 2.5, and 6 mm) is maintained at a constant rate of 20 r/min. Under these conditions, a core-shell vascular graft (CS1508) featuring a PLGA core and PCL shell was fabricated. Additionally, pure PCL fiber tubes and membranes were produced via electrospinning under identical parameters. After electrospinning, the vascular graft (CS1508) or fiber membrane was dried in a fume hood for 4 h, transferred to a vacuum drying oven at 37 ℃ for 24 h, and then refrigerated in an airtight glass tube.

### Surface heparinization of PLGA@PCL SDVGs

The heparinization of PLGA@PCL SDVGs were carried out according to our previous work [18,21]. Firstly, the PLGA@PCL SDVGs were corroded in 0.1 mol·L⁻¹ NaOH solution for 30 h to obtain -COO-. Secondly, these grafts were immersed into a solution of EDCI and DIPEA in MeOH in an ice bath for 15 min, and HOBT was added keeping in the ice bath for further 2 h. And then, H2N-PEG-NH2 was then added into the MeOH solution and reacted at room temperature for 24 hours. Finally, EDCI and NHS were added to the MES solution containing heparin and stirred in an ice-water bath for

2 h. Subsequently, the PLGA@PCL SDVGs obtained in the previous step were put into the reaction solution and inverted in a shaker at 37 ℃ for 24 h. And then, the reacted grafts were rinsed with deionized water and graded ethanol before vacuum-dried. The heparinized PLGA@PCL core-shell structure fiber SDVGs (Hereinafter referred to as "HCS1508 SDVGs") were then kept in the refrigerator and away from light.

## Physicochemical characterization of the PLGA@PCL SDVGs

In the electrospinning process, a copper mesh was used to collect test samples before the receiver, which were then dried in a fume hood for 4 hours, and then placed in a vacuum drying oven for 24 hours. The structure of the fibers was observed using transmission electron microscopy (TEM, JEM-1200EX, Japan) at 100 kV. The vascular grafts were then cut into pieces of 3 mm and sprayed with gold and the surfaces of all the samples were observed with a scanning electron microscope (SEM, Hitachi S-4800, Japan). A total of 100 fibers were randomly selected from each group, and the diameter of grafts' fibers was measured by Image-Pro Plus 6.0 (Media Cybernetics, US).

## Study on the mechanical properties of the PLGA@PCL SDVGs

Vascular graft with the diameter of 5 mm and the thickness of 3 mm were cut into rings with the length of 4 mm. And then the tensile properties were measured using an electronic universal tensile machine (DXLL-5000, Shanghai, China) with the tensile speed of 2 mm/min. Vascular grafts with the diameter of 5 mm, the thickness of 3 mm, and length of 6 cm were used to test the burst pressure. The burst pressure of the samples was measured using a combination device of a syringe pump and a pressure gauge. The graft was securely attached to one end of a connecting tube, which was then connected to an injection pump and a pressure gauge. The other end of the graft was neatly sealed with a tie. PBS was injected into the graft through the syringe pump until the rupture of the graft. The maximum value of the pressure gauge was recorded and repeated at least 3 times for each sample.

## Study on the in vitro degradation of the vascular grafts

The dried vascular grafts were immersed in PBS (Phosphate buffer saline) solution of type I collagenase (5 U/ml) in a water bath shaker at 37 ℃ and 80 r/min. The type I collagenase solution was changed every 7 days. The degradation of the PLGA@PCL grafts was observed after 1, 3, 5 and 7 weeks. At each time point, the samples were rinsed 4 times with deionized water, and each cleaning was completed in a 40 rpm room temperature oscillator. Finally, the samples were dried in a freeze-dryer for 16 h. The weight loss ration was calculated according to the following formula, and the molecular weight changes of the vascular grafts after degradation *in vitro* were tested by gel permeation chromatography (GPC) [22].

$$\text{Weight loss ratio} = 100\% * (w_0 - w_t) / w_0$$

where $w_0$ is the mass before degradation, $w_t$ is the mass after accelerated degradation.

## Study on the release of heparin

The toluidine blue assay was used to determine the release of heparin from the grafts. Briefly, graded standard heparin (0, 5, 10, 15, 20, 25, and 30 μg/ml) solutions were prepared with PBS. The HCS1508 SDVGs of the determined area were immersed in a 15 ml centrifuge tube of 2 ml PBS followed by 3 ml toluidine blue solution. The centrifuge tube was then placed in a constant temperature water bath at 37 ℃, incubate at 40 rpm for 30 min, and then add 3 ml of normal hexane, oscillate with a vortex oscillator for 2 min, and then leave at room temperature for 10 min. The absorbance of each sample and the standard solution at 631 nm was measured. The relationship between heparin content and absorbance was determined according to the absorbance curve of the established standard solution, so as to calculate the heparin content per unit area of the vascular grafts after heparinization. Each group of samples was repeated 3 times [19].

 

## Heparin activity test

The anticoagulation activity of HCS1508 SDVGs was measured by activated partial thromboplastin time (APTT) assay using a semi-automatic coagulation analyzer (TEChrom IV Plus, China). Human plasma was used to finish this test. Briefly, HCS1508 SDVGs were fixed in the bottom of the sample pool, 50 µL of plasma and 50 µL of ellagic acid working solution were added and incubated at 37 °C for 3 minutes. After that 50 µL of $CaCl_2$ solution was added, the clotting time was recorded and each group of samples was measured for 4 times.

## Study on the effectiveness of vascular replacement in rabbits

**Surgical procedures.** 14 New Zealand white rabbits weighting 2.8–3.2 kg were used in this study. All animal experiments performed for this study were approved by the Experimental Animal Ethics Committee of Fuwai Hospital (IACUC approval number 0090-4-45-HX(Z)) and Kangtai Medical Laboratory Service Hebei Co., LTD (IACUC approval number MDL2024-12-04-02). The preparation work before was conducted as follows: The HCS1508 and pure PCL SDVGs were separately immersed in 75% alcohol for a duration of 20 minutes, and then rinsed with sterile saline solution for three times. Anesthesia induction of the rabbits was achieved using propofol (5–8 mg/kg), followed by maintenance of anesthesia through the administration of 2% isoflurane after endotracheal intubation. Systemic anticoagulation of the animals was accomplished via intravenous injection of low molecular weight heparin (100 IU/kg).

For a formal operation, an incision was made on the neck skin and right muscle to expose the right carotid artery initially. The proximal and distal ends of the carotid artery were blocked with vascular clamps, and the carotid artery was cut between the two clips. Saline solution was utilized to flush out any blood from the anastomosis and the outer membrane of the anastomosis were removed by the microsurgical scissors. Subsequently, 7−0 sutures were used to perform end-to-end anastomosis between the vascular graft (inner diameter 2.5 mm, length 2.0 cm) and the rabbit carotid artery by intermittent suture. Finally, the artery clamps were slowly removed to restore blood flow and the medical gauze was applied to the sutured site to stop the bleeding. 3−0 sutures were used to close the muscle layer and skin of the neck. Additionally, the rabbits were given buprenorphine (0.05 mg/kg) and meloxicam (0.3 mg/kg) after surgery. Within 3 days after implantation, 2000 IU of heparin sodium was injected subcutaneously every day. Warfarin of 0.75 mg/d were given orally for 1 month.

**Color Doppler ultrasonography and sampling of vascular grafts.** Color Doppler ultrasound (Vevo 2100, VisualSonics, Canada) was used to observe the patency, blood flow and the diameter of the SDVGs after the anesthesia of the rabbits. The animals in the HCS1508 SDVGs group were euthanized at 3, 6 and 9 months respectively, and the animals in the PCL group were euthanized at 3 months.

**Histological analysis of the implanted vascular grafts.** The rabbits in the HCS1508 SDVGs group were euthanized by intravenous injection of 3% pentobarbital sodium at 3, 6 and 9 months. The rabbits in the PCL group were euthanized at 3 months. Subsequently, the implanted grafts were collected. And then, these grafts were dehydrated in gradient concentration ethanol after fixation with formalin and embedded in paraffin. The cross-sectional slices of the vascular grafts were collected for hematoxylin & eosin (H&E) staining. Immunofluorescence staining was conducted using anti-CD31 antibody (Abcam, USA) for the testing of endothelial cells, anti-α-SMA and antibody and anti-MHC (Abcam, USA) for SMCs, anti-CD68 antibody (Abcam, USA) for macrophages. The slides were examined under a fluorescence microscope (Olmpus, DS-U3, Japan) and images were captured. The histological characteristics and the tissue regeneration of the implanted vascular grafts were evaluated.

**Statistical methods.** All of the data were expressed as mean value±standard deviation (SD) for each group. The results were analyzed using the software SPSS 26.0, performing T-test to determine the statistical significance. P-values less than 0.05 were considered statistically significant and * < 0.05, ** < 0.01, *** < 0.001.

## Results and discussion

### Physicochemical characterization of vascular grafts

As showed in Fig 1, the diameter of CS1508, pure PCL and HSC grafts were 3.20±0.66μm, 5.01±1.30μm and 2.69±0.71μm, respectively. All of the SDVGs showed continuous and smooth fibrous morphology on the inner surface. TEM images further showed that the vascular grafts maintained a good core-shell structure under the electrospinning procession.

### Study on the mechanical properties of the vascular grafts

The Young's modulus of CS1508, PCL and HCS1508 were 9.40±1.23MPa, 8.27±1.65MPa and 9.70±0.62MPa, respectively. The elongation at break of these three materials were 592.33±27.82%, 538.15±74.30%, 536.78±38.48%, and the maximum tensile strength were 2.42±0.14MPa, 1.28±0.50MPa, 2.82±0.71MPa, separately (Fig 2A). The results showed that the maximum tensile strength of the core-shell vascular grafts were higher than that of the PCL grafts, which was higher than that of the human aphenous vein (~1.8MPa). The burst pressure of CS1508, PCL and HCS1508 SDVGs was 1140.09±99.01 mmHg, 1267.60±102.76 mmHg, 1170.10±72.75 mmHg, respectively (Fig 2B). These values are 7–9 times higher than men's natural blood pressure of 120 mmHg.

### Study on the in vitro degradation of the vascular grafts

After 7weeks of accelerated degradation in type I collagenase, the mass loss rate of CS1508, PCL and HCS1508 were 5.44±0.48%, 2.48±0.22% and 6.24±0.83%, respectively (Fig 2C). The GPC molecular weight of the samples before and after degradation was measured, and the results showed that the molecular weight of all samples decreased to different degrees after 7 weeks of degradation (Fig 2D). The molecular weight of pure PCL decreased the least compared with

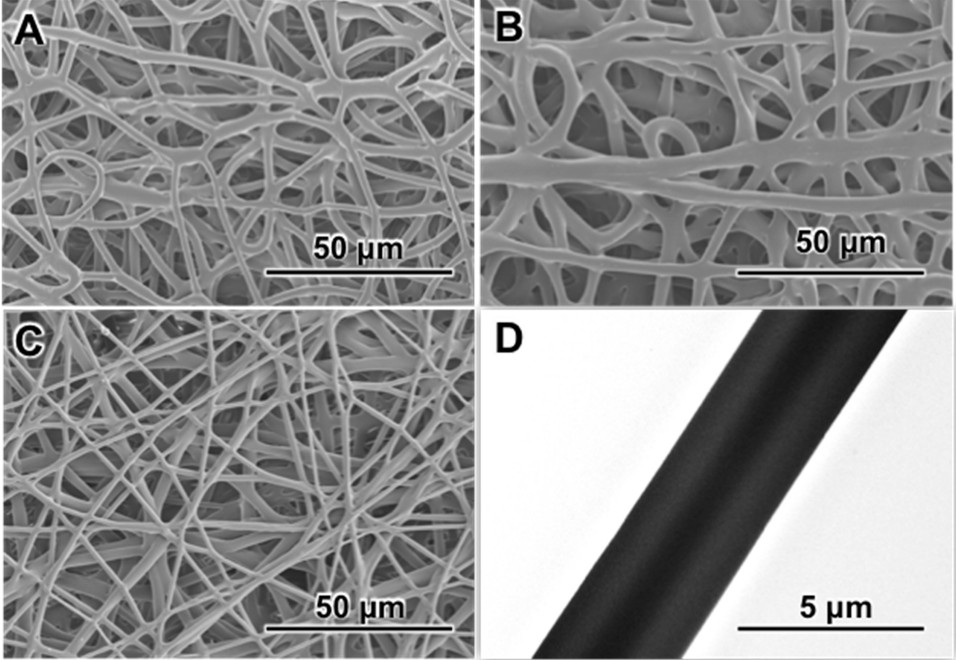

**Fig 1. Surface morphology of the fabricated vascular grafts.** (A) CS1508 (B) PLA (C) HCS1508 (D) TEM of CS1508. The scale bar in (A-C) is 50 μm and the scale bar in Figure D is 5 μm.

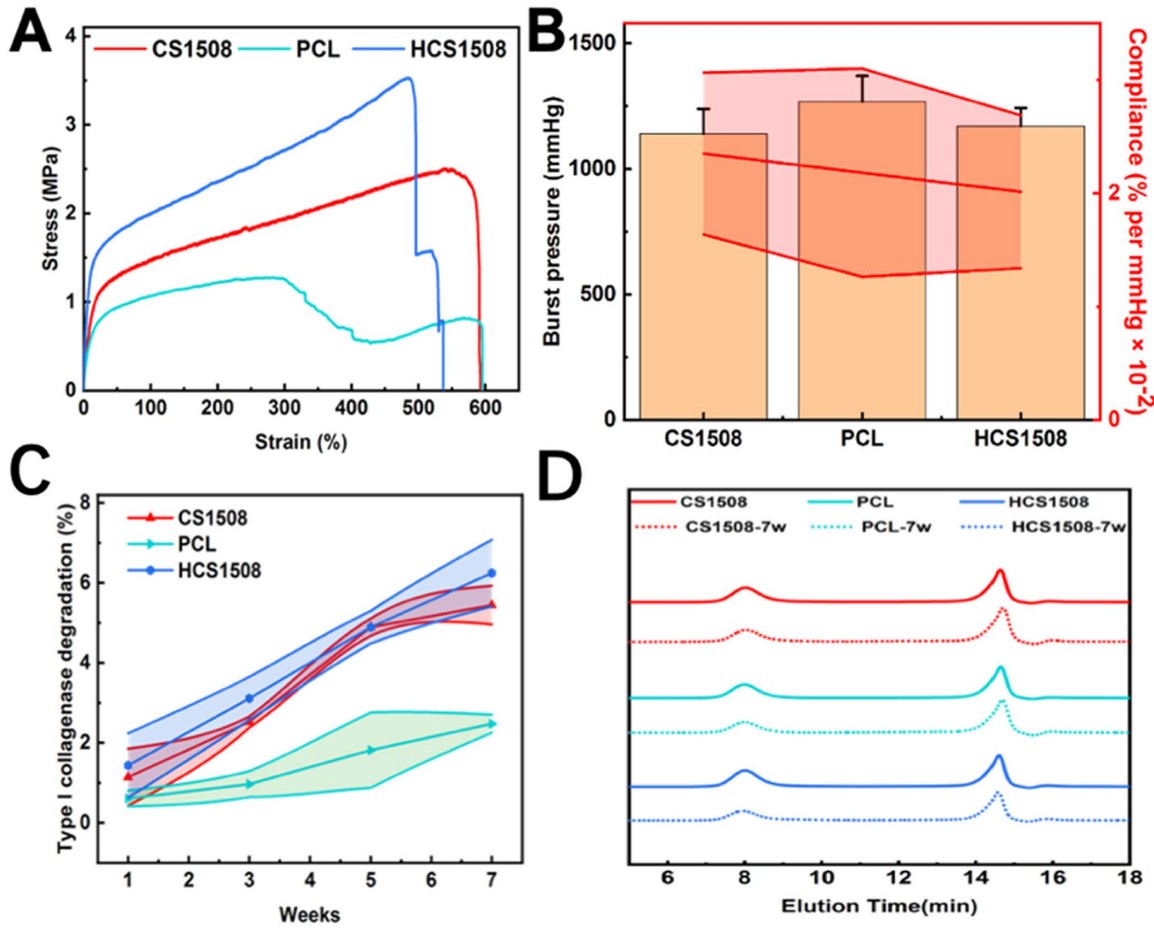

**Fig 2. The results of mechanical analysis and *in vitro* degradation tests of the SDVGs of CS1508, PCL and HCS1508.** (A) Stress-strain curves of typical mechanics. (B) Compliance and burst pressure. (C) Weight loss ratio of *in vitro* degradation test. (D) GPC results of *in vitro* degradation test.

the SDVGs of CS1508 and HCS1508, which may be related to the fact that PLGA was exposed from the core layer of the fiber during the accelerated degradation process and participated in the degradation process, resulting in an acidic environment that accelerated the degradation of the material. The pure PCL maintained a relatively neutral environment during the degradation process, so the molecular weight did not change significantly. It can be seen from the mass loss rate and molecular weight test that the mass loss of vascular graft with core-shell structure is greater than that of pure PCL material, indicating that the preparation of core-shell structure can accelerate the degradation of materials *in vivo* and provide more space for cells, which is beneficial to the regeneration of vascular tissue.

## Study on the release of heparin

The surface heparin content of HCS1508 SDVGs was about 13.74 µg/cm² by toluidine blue staining (Fig 3A), and the result of APTT showed that the coagulation time was extended by 16.8±4.7s compared with normal plasma. At the same time, heparin release of HCS1508 SDVGs in PBS buffers was tested for 28 days. The release rate of surface heparin in HCS1508 SDVGs showed the highest at the first week, about 2.44% per day, and then gradually decreased at the following three weeks, about 0.71% per day. After 28 days of *in vitro* release, the cumulative heparin release of HCS1508 was

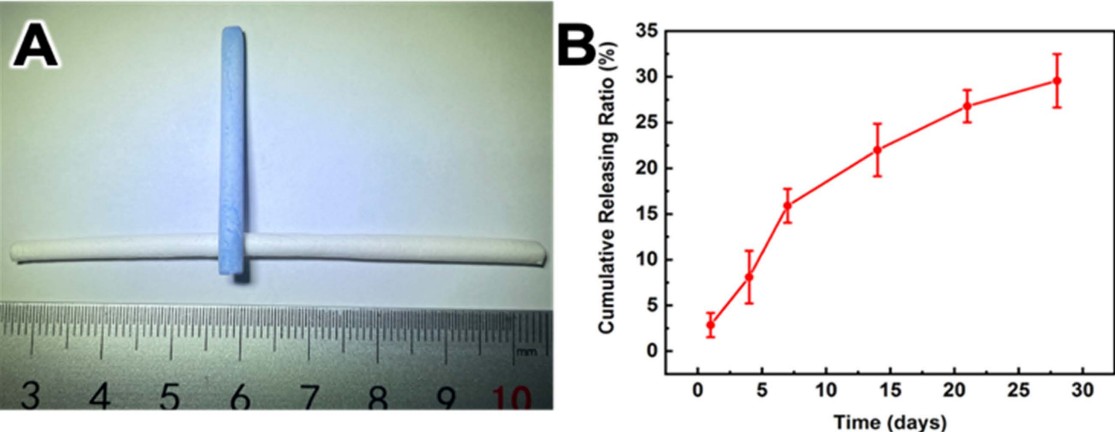

**Fig 3. The release of heparin different proportions.** (A) The specific reaction of HCS1508 with toluidine blue after heparinization (B) *in vitro* release of HCS1508.

32.04±2.61% (Fig 3B). These results demonstrated that the core-shell structure could be used to achieve the gradual release of heparin.

### Study on the effectiveness of vascular replacement in rabbits

All the rabbits were alive during and after operation. Except for one obvious vascular stenosis in the second month after implantation (stenosis rate was 70.1%, Fig 4A), no other vascular stenosis or aneurysm occurred in the rabbits during the whole observation period and the implanted HCS1508 grafts showed a high vascular patency (100%). In addition, one of the four pure PCL vascular grafts showed vessel occlusion at 3 month, while the remaining three had smooth luminal surfaces. Rabbits in PCL group were euthanized at 3 month. (Fig 5).

H&E staining showed that after carotid artery replacement of HCS1508 grafts, the thickness of the intima was only 28 μm at 3 months, but reached 218 μm at 9 months after implantation, increased by a factor of 7.8. The Von Kossa staining of HCS vascular grafts showed that calcification occurred in the vascular grafts from 3 months, and calcification mainly occurs in the cell aggregation area in the grafts' wall which also happens in the pure PCL grafts (Fig 7). The calcification area of HCS1508 in the 3-month and 6-month postoperative samples was closer to 3.5±0.8% and 4.2±1.1%, respectively, whereas at 9 months the area of calcification was only 1.4±0.6% of the grafts. Though the calcification area of pure PCL grafts reached 0.2±0.1% after 3 months *in vivo*, which were more significant low than those HCS1508 grafts for less cell aggregation. The results of Masson staining further demonstrated that the contents of collagen showed a trend of first decreasing and then increasing with the extension of the implantation time of HCS1508 SDVGs *in vivo*. The distribution of collagen showed the trend of uniform from the implantation of 3 months, which was more similar with the autogenous blood vessels. However, there was no significant difference in elastin content after 3 months (Fig 6). As for pure PCL grafts, after 3 months, the regeneration of collagen and elastin distributes across the wall of the grafts for a looser con-structure, and the content of collagen was lower than that of HCS1508 grafts (Fig 8). As a contrast, the content of elastin was significantly higher than HCS1508 grafts (Fig 7 and Fig 8).

Immunohistochemical staining was used to specifically stain vascular smooth muscle cells (VSMCs) regeneration and contractile VSMCs expression during transplantation. It was observed that SMCs mainly distributed in the intima layer of vascular regeneration in the medial lumen during the early implantation of HCS1508 vascular graft. With the extension of transplantation time, VSMCs gradually began to enter the graft blood vessels and showed scattered distribution in the tube wall. By 9 months after transplantation, the expression of VSMCs was abundant on the

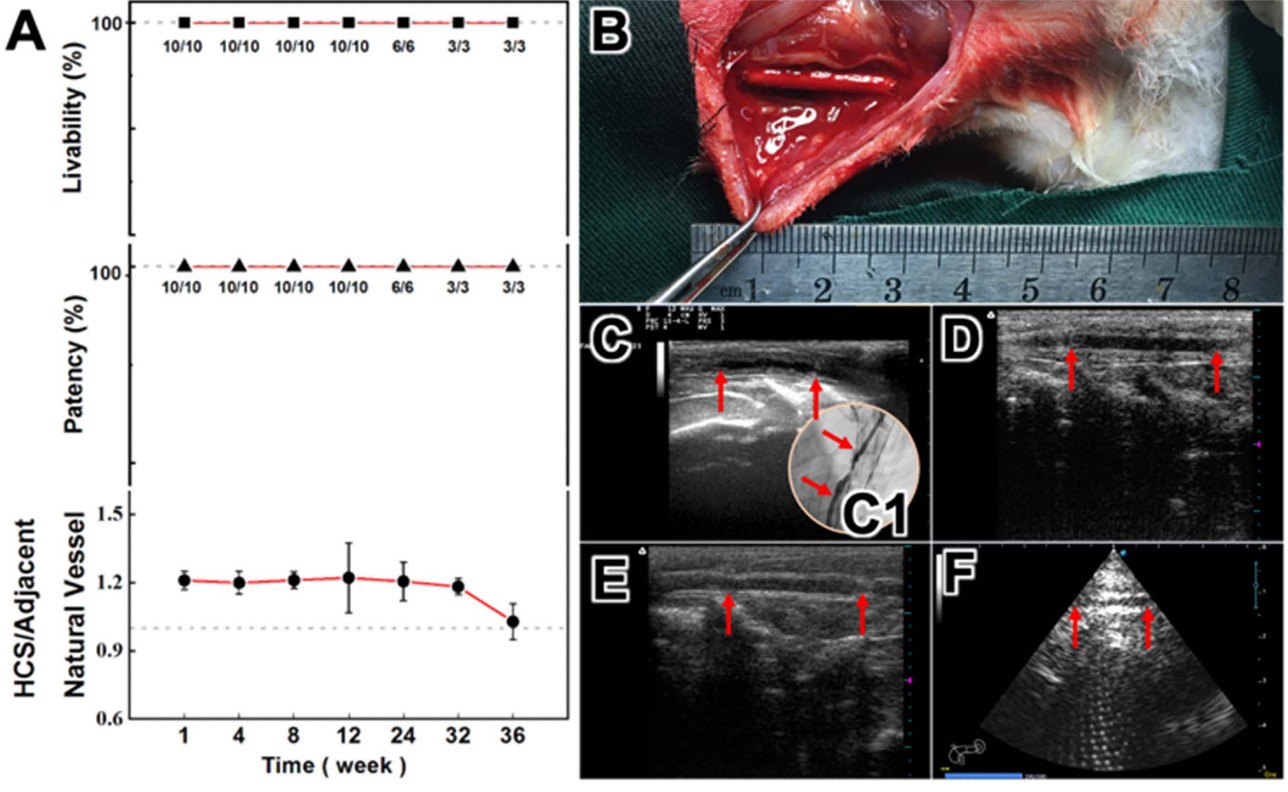

**Fig 4. The results of gross observation.** (A) Livability, patency and vessel diameter changes of HCS/ANV (adjacent natural vessel) in New Zealand White rabbits. (B) Surgical grafting procedure. (C-F) Typical ultrasound images of HCS1508 SDVGs at 2, 3, 6, and 9 months after surgery, respectively, with HCS1508 SDVGs in between the two red arrows.

surface of the internal and external lumen of the grafted vessels, reaching 41.9% of that of the autologous SMCs. The expression distribution of MHC-labeled extensional VSMCs also showed a similar pattern, and the expression level was even as high as 159.2% of that of self at 9 months (Fig 9). At 3-month, the content of regenerated VSMCs in pure PCL vascular grafts (~2.18%) was basically the same. But VSMCs mainly located in the initial and external surface of the pure PCL grafts which is different from that in HCS1508 grafts. As for the contractile VSCMs, the content of pure PCL grafts was higher than HCS1508, which could be due to the high expression of macrophage in pure PCL grafts (Fig 10).

The results of CD31 immunohistochemical staining showed that the HCS1508 SDVGs expressed shallow positive areas on the internal surface of the lumen at 3 months after implantation, and the positive areas inside the lumen were significantly darker at 9 months (A1-A4 in Fig 10). In other words, the HCS1508 showed relative complete endothelialization after implantation for 3 month and the quantity of the vascular endothelial cells increased at 9 months. These vascular endothelial cells formed a tight arrangement and the cell density gradually approached or even equated to the ECs staining results of autologous blood vessels. As for pure PCL grafts, 3 months later, the endothelialization was also not completely (C1 in Fig 11). The results of CD68 staining further showed that in rabbits, the main expression area of HCS1508 vascular graft was in the vascular wall and the outer surface of the vascular graft after implantation for 3 months which was similar with the pure PCL grafts in rabbits (B1-B4 in Fig 10 and Fig 11D), while there were multiple layers of irregular distribution in the blood vessels of the graft at 9 months. The content of CD68-labeled macrophages showed a trend of increase after the light density calculation of CD68 staining.

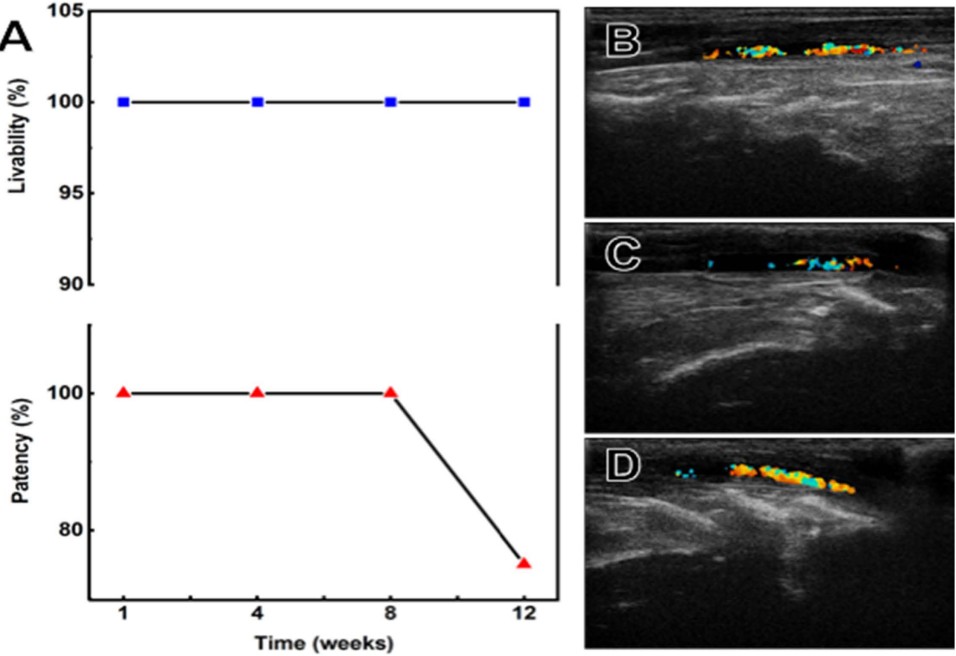

**Fig 5. Results of PCL grafts following 3 months of implantation. (A)** Livability and Patency of pure PCL grafts in New Zealand White rabbits. **(B-D)** Results of 3-month ultrasonography of pure PCL vascular grafts.

## Discussion

The small-diameter artificial vascular graft should have good blood compatibility, tissue compatibility and mechanical properties. At the same time, rapid endothelialization of the vascular lumen in a sufficient range is the main condition for maintaining long-term patency of the vascular graft [6]. PCL has emerged as a widely utilized synthetic vascular material in the fabrication of SDVG, owing to its excellent plasticity, biodegradability, and favorable mechanical properties [23]. However, the properties of PCL such as hydrophobic properties, poor cell adhesion and affinity limit their clinical application. PLGA is a copolymer of ethyl and lactide, which usually has different physicochemical and biological properties according to the proportion of pre-polymerization monomers [24]. The degradation time of 50/50 PLGA is about 1–3 months. The former experiment has verified that HCS1508 could basically match the mechanical properties of human saphenous vein and faster degradation than CS1508 *in vitro*. Also the rat abdominal aortic replacement model demonstrated its patency rate of 88.9% and more rapid endothelialization ability. Hererin, the same strategie was used to fabricate PLGA@PCL nanofiber SDVGs and used as carotid vascular access in rabbits for a more beneficial effect of cell infiltration and vascular tissue regeneration [20].

The formation of an endothelial layer with a complete architecture and function can effectively reduce the formation of thrombus, inhibit the proliferation of VSMCs, and control blood flow interactions with the vessel wall, which is crucial to enhance long-term patency. Up to now, various strategies have been continuously tried to optimize the endothelialization of small-diameter artificial vascular grafts[Improving Vascular Regeneration Performance of Electrospun Poly(ε-Caprolactone) Vascular Grafts via Synergistic Functionalization with VE-Cadherin/VEGF、 Rapid endothelialization of small diameter vascular grafts by a bioactive integrin-binding ligand specifically targeting endothelial progenitor cells and endothelial cells、 Ethanol-lubricated expanded-polytetrafluoroethylene vascular grafts loaded with eggshell membrane extract and heparin for rapid endothelialization and anticoagulation]. In the present study, the patency of the HCS1508 vascular grafts was kept at 100% after 3, 6 and 9 months. In this study, HCS1508 grafts showed relatively complete endothelialization

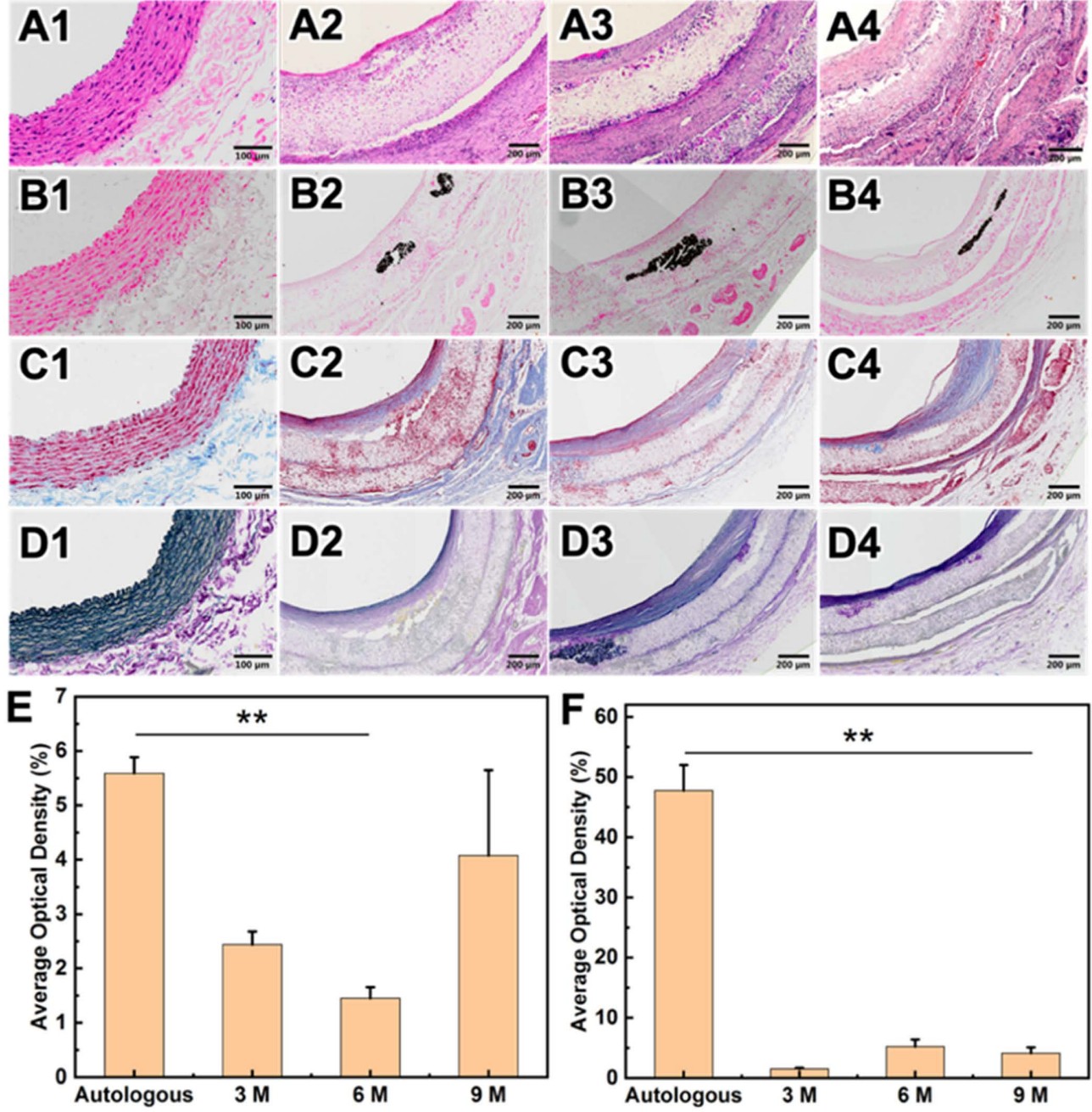

**Fig 6. Histological staining results of the native blood vessel and the HCS1508 SDVGs implanted after 3, 6 and 9 months.** (A1-A4) H&E staining. (B1-B4) Von Kossa staining. (C1-C4) Masson staining. (D1-D4) EVG staining results of autologous blood vessels and vascular grafts at 3, 6, and 9 months after transplant into rabbit. (E) Average optical density results for collagen. (E) Average optical density results for elastin.

at 3 months after implantation, and endothelial cells were tightly arranged on the surface of the lumen at 9 months, and the cell density was even close to that of autologous blood vessels. This formation of the inner cortex further reduced the occurrence of coagulation, inhibited the proliferation of vascular intima, and ensured the long-term patency of blood vessels.

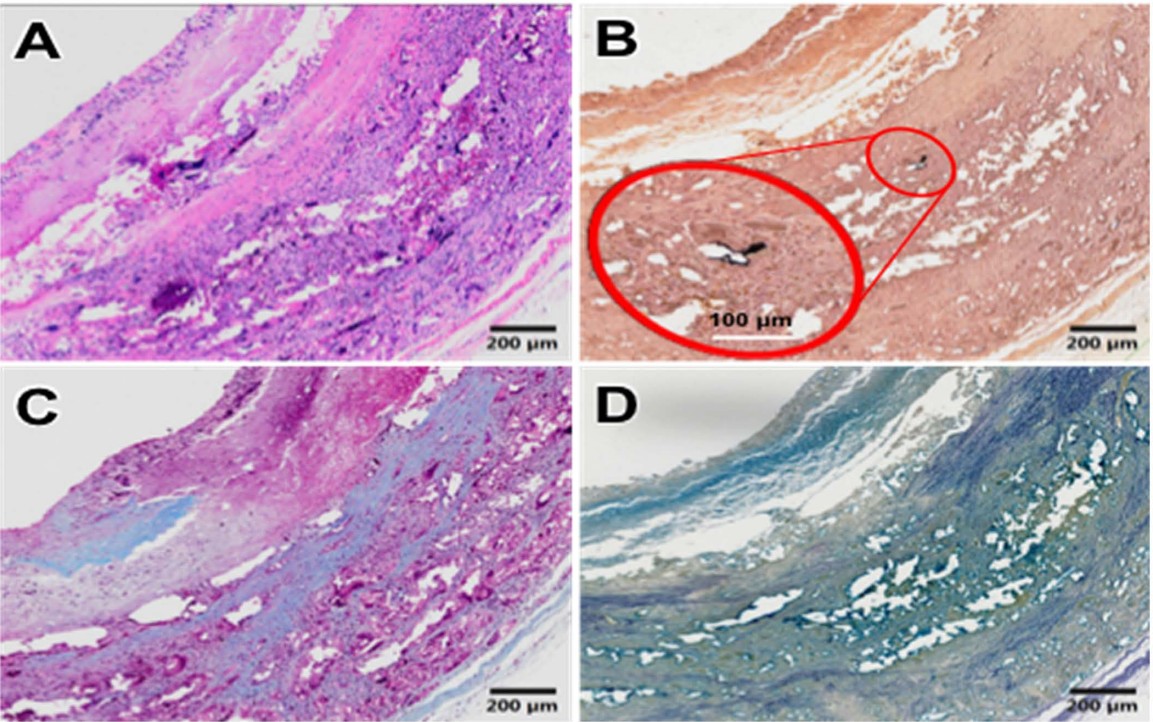

**Fig 7. Tissue staining results of PCL vascular grafts at 3 months.** (A) H&E staining, (B) Von kossa staining, (C) Masson staining, (D) EVG staining.

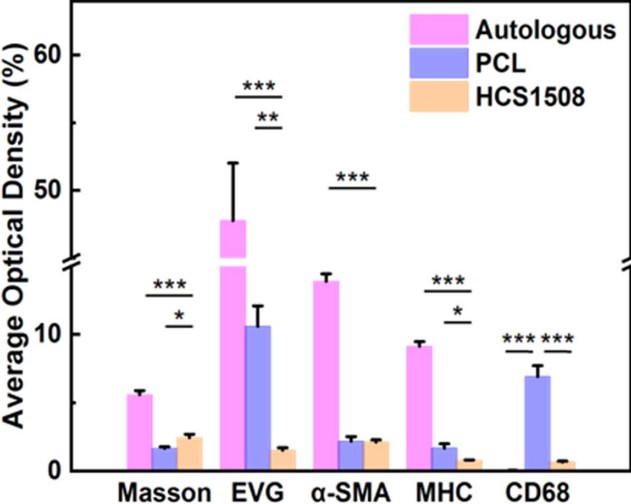

**Fig 8. Average optical density of different staining results of PCL vascular grafts at 3 months.**

VSMCs are the most abundant cells in the blood vessels and located in the media layer. These cells have significant plasticity, sensing, adapting and influencing other cell types and their environment. The ability of VSMCs to transition from a static "contractile" phenotype to a proliferative "synthetic" phenotype is crucial for vascular injury repair

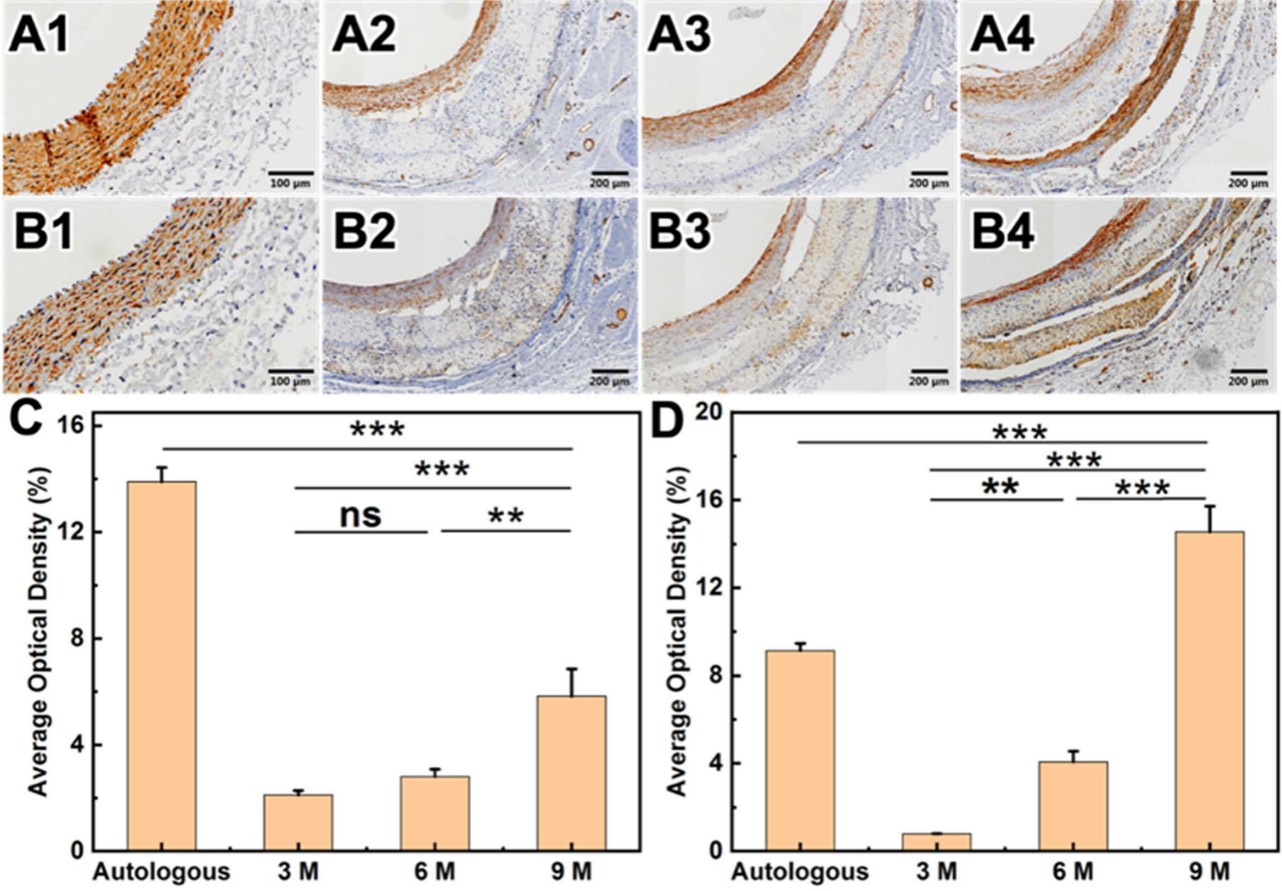

**Fig 9. Results of immunohistochemical staining of HCS1508 SDVGs in rabbit carotid artery autologous blood vessels and postoperative 3, 6 and 9 months.** (A1-A4) Results of α-SMA staining. (B1-B4) Results of MHC staning. (C) Average optical density results of α-SMA staining, (D) Average optical density results of MHC staining.

[25]. It can be seen that with the extension of time, the number of VSMCs gradually increased, and the distribution position changed from only in the medial lumen to the internal and external lumen. At the same time, the MHC-labeled contractile VSMCs also showed a similar change pattern, while the MHC can label the contractile VSMCs, which can participate in the multiple functions of SMCs for the regulation of vascular homeostasis. These results were also conducive to vascular remodeling. These results also proved that HCS1508 vascular grafts were beneficial to vascular remodeling.

As a foreign material, the first process of HCS1508 vascular graft regeneration *in vivo* was accompanied by the occurrence of foreign body reaction. The proliferation of inflammatory cells in the early stage greatly increased the secretion of extracellular matrix. With the decrease of inflammatory response, the secretion of extracellular matrix also showed a decrease trend. With the degradation of materials *in vivo*, the proliferation of VSCMs gradually occurred, and finally the content of collagen showed a rising trend again. The results of CD68 staining showed that macrophages infiltrated from the outer layer to the inner layer of the vascular grafts and the quantity were gradually increased. This was mainly consistent with the progressive degradation of materials and the phenotypic function of macrophages [26,27]. Although pure PCL vascular grafts expressed a higher content of VSMCs with a telescopic phenotype at 3 months, the expression of inflammatory cells during vascular remodeling was much higher than that of HCS1508 The hydrophilic modification of

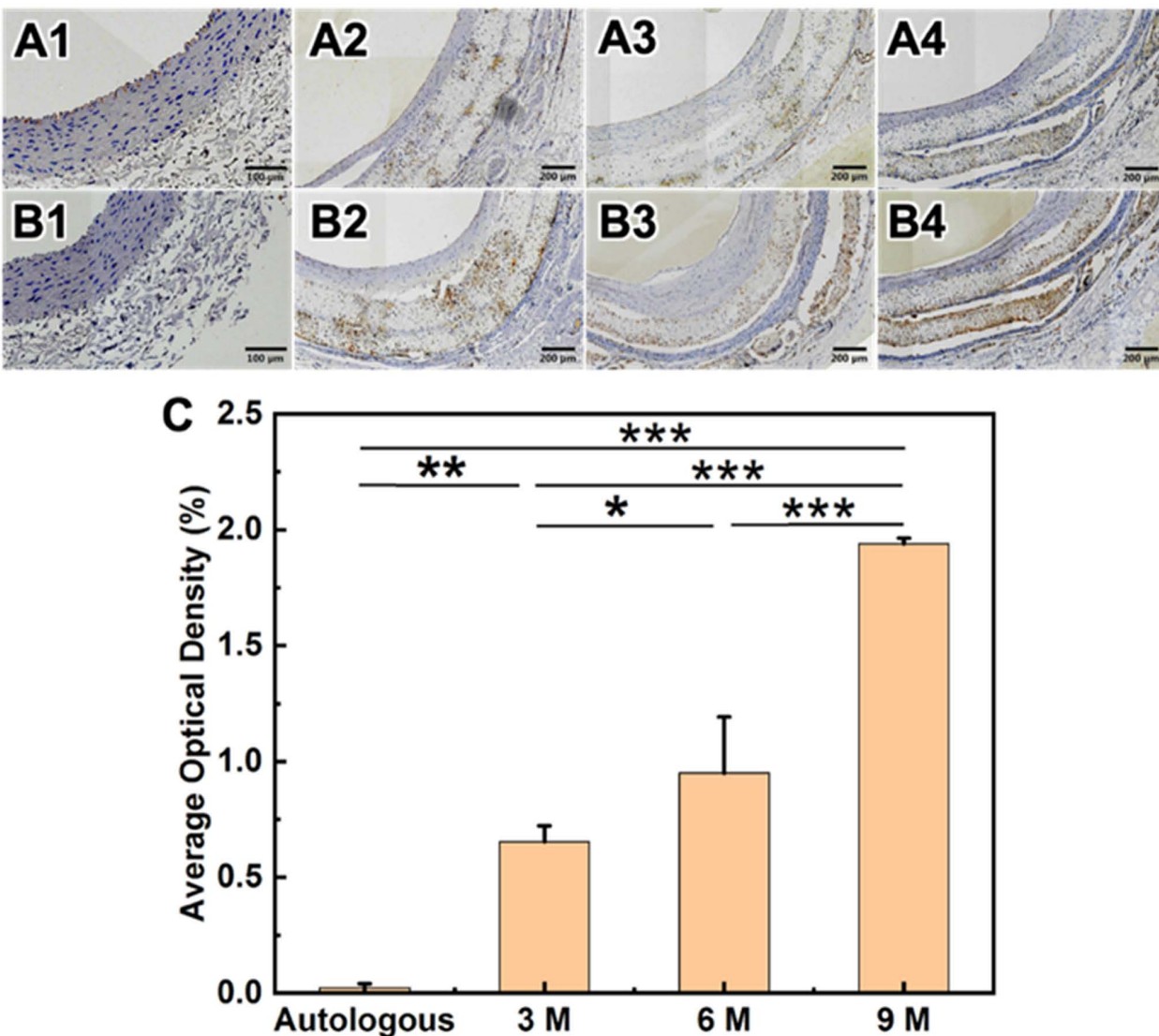

**Fig 10. Results of immunohistochemical staining of HCS vascular grafts in rabbit carotid artery autologous blood vessels and postoperative 3, 6 and 9 months.** (A1-A4) CD31 staining results. (B1-B4) CD68 staining results (C) average optical density results of CD68 staining.

heparin was able to gently influence the process of vascular remodeling *in vivo*, thereby reducing the potential for vascular aneurysm development during remodeling.

It is imperative to acknowledge that this study does possess certain limitations. Firstly, the rabbit carotid artery transplantation model was employed to assess the efficacy of in vivo repair of small-diameter vascular grafts (SDVGs). Nevertheless, a significant disparity exists between the mechanisms of vascular remodeling observed in rabbits and those occurring in humans [28]. It is important to note that the formation of intima in rabbits is primarily attributed to endothelial migration through anastomosis, which differs from the main process of endothelization in human beings [29]. Additionally, Calcification sites were observed at 3, 6, and 9 months after the implantation of HCS1508 in rabbits, which may lead to brittle or ruptured graft, thereby leading to the decline and loss of its intended function and shortening the expected

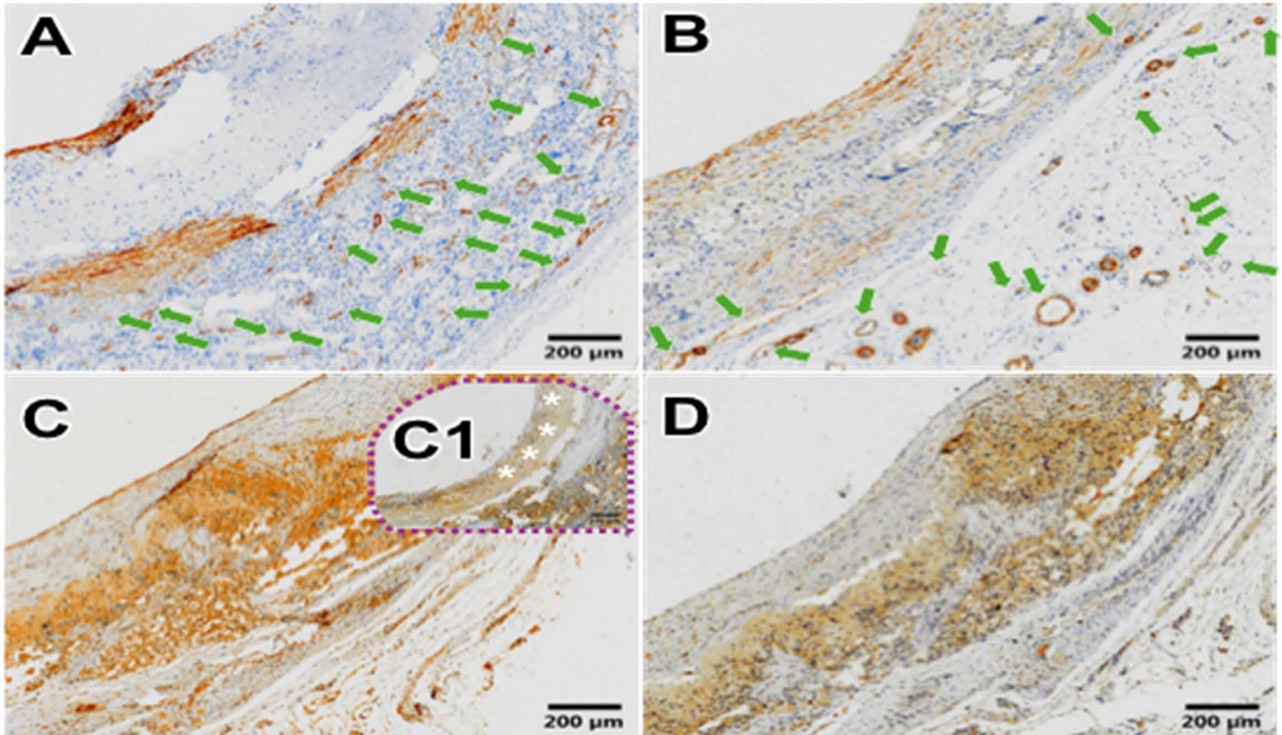

**Fig 11. Immunohistochemical staining results of PCL vascular grafts at 3 months.** (A) α-SMA staining, (B) MHC staining, (C) CD31 staining, (D) CD68 staining. Neovascularization of the neovascular tissue was shown by green arrows.

service life of the material [30,31]. The improvement of anti-calcification performance will definitely be a focus for the fabrication of small vascular grafts in the future.

## Conclusions

In conclusion, the PLGA@PCL core-shell structural fibrous SDVGs with surface heparinized were fabricated in this study. The results indicate that the heparin-modified PLGA@PCL core-shell structural fibrous SDVGs achieved long-term patency and the arrangement of collagen and elastin in the neointima was similar to the native vessel in the rabbits after 9 months. After 3 months postoperatively, endothelialization was almost complete, and vascular calcification was also observed. It can be concluded that surface heparinization is a feasible modification method for in situ tissue-engineered vascular grafts, and controlling the occurrence of vascular calcification is another important issue to be solved in the development of SDVGs, and it is also the focus of our next research work.

## Acknowledgments

We would thank Chunren Wang from National Institute for Food and Drug Control, Beijing, China, for his support and guidance to this research.

## Author contributions

**Conceptualization:** han wang, Yonghao Xiao.

**Formal analysis:** Yonghao Xiao.

**Funding acquisition:** han wang, Zengguo Feng, Hongbo Chen, Yongquan Gu.

**Investigation:** Yonghao Xiao, Lin Ye, Ai-ying Zhang, Xue Geng, Fanshan Qiu.

**Methodology:** Yonghao Xiao.

**Project administration:** Yongquan Gu.

**Resources:** han wang, Yuhao Jiao, Yuehao Xing.

**Supervision:** Zengguo Feng.

**Validation:** han wang, Lin Ye.

**Visualization:** Yonghao Xiao.

**Writing – original draft:** han wang.

**Writing – review & editing:** Yonghao Xiao.

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
