## [Decision Letter · Decision Letter 0]

12 Aug 2024

Dear Dr. wang,

Thank you for submitting your manuscript to PLOS ONE. After careful consideration, we feel that it has merit but does not fully meet PLOS ONE’s publication criteria as it currently stands. Therefore, we invite you to submit a revised version of the manuscript that addresses the points raised during the review process.

as per the comments below from Reviewer 1 and reviewer 2.

We look forward to receiving your revised manuscript.

Kind regards,

Pradeep Kumar, Ph.D.

Academic Editor

PLOS ONE

Journal Requirements:

Fabrication and performance evaluation of PLCL-hCOLIII small-diameter vascular grafts crosslinked with procyanidins

Author links open overlay panel - https://doi.org/10.1016/j.ijbiomac.2023.126293

Remodeling of structurally reinforced (TPU+PCL/PCL)-Hep electrospun small-diameter bilayer vascular grafts interposed in rat abdominal aortas - https://doi.org/10.1039/D1BM01653A

(Among others)

In your revision ensure you cite all your sources (including your own works), and quote or rephrase any duplicated text outside the methods section. Further consideration is dependent on these concerns being addressed.

 [This work was supported by the National Key R&D Program of China (No.2022YFC2409802) and the National Key R&D Program of China (2017YFC1104101).].  

[This work was supported by the National Key R&D Program of China (No.2022YFC2409802) and the National Key R&D Program of China (2017YFC1104101).]

 [This work was supported by the National Key R&D Program of China (No.2022YFC2409802) and the National Key R&D Program of China (2017YFC1104101).]

6. In the online submission form, you indicated that [Data can be obtained by emailing the corresponding author of this article.]. 

8. We note that Figure(s) 1, 3a and 4 in your submission contain copyrighted images. All PLOS content is published under the Creative Commons Attribution License (CC BY 4.0), which means that the manuscript, images, and Supporting Information files will be freely available online, and any third party is permitted to access, download, copy, distribute, and use these materials in any way, even commercially, with proper attribution. For more information, see our copyright guidelines: http://journals.plos.org/plosone/s/licenses-and-copyright.

a. You may seek permission from the original copyright holder of Figure(s) 1, 3a and 4 to publish the content specifically under the CC BY 4.0 license. 

Reviewers' comments:

Reviewer's Responses to Questions

**Comments to the Author**

1. Is the manuscript technically sound, and do the data support the conclusions?

Reviewer #1: Yes

Reviewer #2: Yes

2. Has the statistical analysis been performed appropriately and rigorously?

Reviewer #1: No

Reviewer #2: Yes

3. Have the authors made all data underlying the findings in their manuscript fully available?

Reviewer #1: No

Reviewer #2: Yes

4. Is the manuscript presented in an intelligible fashion and written in standard English?

Reviewer #1: Yes

Reviewer #2: Yes

Reviewer #1: In this study, a small-diameter artificial blood vessel was fabricated using PLGA@PCL core-shell fiber, and by immobilizing heparin on the surface, a highly patent graft was successfully obtained. Interestingly, the authors evaluated the data from material evaluation to in vivo experiments. On the other hand, many reports have described artificial blood vessels fabricated by electrospinning for a long time. For this reason, it is necessary to clarify the unique characteristics that are claimed in this study. Our primary comments are summarized below.

1. The definition of Core-Shell fiber, which is the focus of this paper, and its structure should be explained clearly, including schemes.

2. There are no data evaluating the characteristics of Core-Shell fiber, so please add this.

3. In vivo experiments excluded the control experiment. If stable patency rates and tissue regeneration can be induced by blood vessels made of Core-shell fiber, this should be discussed in comparison with the results of other types of fibers, such as those with no core-shell fibers as a control.

Reviewer #2: The manuscript describe electrospinning of PLGA-PCL mat and surface functionalization with heparin.

The electrospinning of PLGA-PCL core-shell mat is well studied area, authors developed small diameter tube out of it for vascular graft application.

The work mainly focus on in-vivo studies and some physicochemical characterization

Though the manuscript is written properly some issues need to addressed before consideration for publication

Manuscript dosent have page and line no

In section 2.1 Materials - sentence "N-Hydroxy succinimide was purchased from Sigma (St. Louis, MO, USA) was purchased from was purchased from Innochem (Beijing, China)" looks incomplete.

In the same section please provide Ethics approval no.

In section 2.2 Fabrication please clearly claim which polymer is forming core and which is forming shell

In section 2.3 authors used many short forms (EDC, DIPEA, HOBT, NHS, MES etc); it might be from their previous publication, please provide full form of the same in this manuscript

In section 2.4 and 2.5 authors should include more characterization like DCS, XRD etc to improve understanding on thermal and crystalline behavior of the scaffolds

In section 2.6 Why author used type I Collagenase for degradation of polymers?

in the same section "wt is the mass before degradation after accelerated degradation for 7 days" looks incomplete

Section 2.8 Please check "werre was..")

Section 2.9.1 authors mentioned that they kept SDVG in 75% alcohol for 20 min. have they studied the effect of alcohol on HCS? please compare it using FTIR

Figure 1 Please provide the cross section of core-shell fiber

Figure 4 (C-D) it should be (C-F)

Just before figure 6 authors mentioned (Figure 8); there is no figure 8 in the manuscript

In discussion please provide full form of VSMC

**Do you want your identity to be public for this peer review?** For information about this choice, including consent withdrawal, please see our Privacy Policy

Reviewer #1: **Yes: ** Atsushi Mahara

Reviewer #2: **Yes: ** Dr. Ravindra Badhe

---

## [Author Response · Author response to Decision Letter 1]

10 Oct 2024

Journal Requirements:

Response: We have modified this paper to meet PLOS ONE's style requirements.

Response: The revised version has included these contents.

Fabrication and performance evaluation of PLCL-hCOLIII small-diameter vascular grafts crosslinked with procyanidins

Author links open overlay panel - https://doi.org/10.1016/j.ijbiomac.2023.126293

Remodeling of structurally reinforced (TPU+PCL/PCL)-Hep electrospun small-diameter bilayer vascular grafts interposed in rat abdominal aortas - https://doi.org/10.1039/D1BM01653A

(Among others)

In your revision ensure you cite all your sources (including your own works), and quote or rephrase any duplicated text outside the methods section. Further consideration is dependent on these concerns being addressed.

Response: We have made modifications to reduce the overlap with our published papers.

[This work was supported by the National Key R&D Program of China (No.2022YFC2409802) and the National Key R&D Program of China (2017YFC1104101).].

Response: We have made modifications in the manuscript and the cover letter.

[This work was supported by the National Key R&D Program of China (No.2022YFC2409802) and the National Key R&D Program of China (2017YFC1104101).]

Please remove any funding-related text from the manuscript and let us know how you would like to update your Funding Statement. Currently, your Funding Statement reads as follows: [This work was supported by the National Key R&D Program of China (No.2022YFC2409802) and the National Key R&D Program of China (2017YFC1104101).]

Response: We have made modifications in the manuscript and the cover letter.

6. In the online submission form, you indicated that [Data can be obtained by emailing the corresponding author of this article.].

Response: All data could be acquired within the manuscript itself.

Response: All experimental procedures were performed under institutional guidelines for animal care and approved by the Animal Ethics Committee of Fuwai Hospital Chinese Academy of Medical Sciences (Beijing, China).

8. We note that Figure(s) 1, 3a and 4 in your submission contain copyrighted images. All PLOS content is published under the Creative Commons Attribution License (CC BY 4.0), which means that the manuscript, images, and Supporting Information files will be freely available online, and any third party is permitted to access, download, copy, distribute, and use these materials in any way, even commercially, with proper attribution. For more information, see our copyright guidelines: http://journals.plos.org/plosone/s/licenses-and-copyright.

a. You may seek permission from the original copyright holder of Figure(s) 1, 3a and 4 to publish the content specifically under the CC BY 4.0 license.

Reviewers' comments:

Reviewer's Responses to Questions

Response: We believe that in the previous version, except picture 1, picture 3 and picture 4 do not involve copyright issues. In the newly revised version, we have also changed picture 1, so currently all images do not involve copyright issues.

Reviewer #1:

1. The definition of Core-Shell fiber, which is the focus of this paper, and its structure should be explained clearly, including schemes.

Response: We have modified the part 2.2 in order to explain the core-shell structure clearly.

2. There are no data evaluating the characteristics of Core-Shell fiber, so please add this.

Response: Thanks for your comment. In this paper, we also characterized the surface morphology and core-shell structure of core-shell fibers by SEM and TEM, and studied the mechanical properties and degradation properties of the vascular graft after surface heparinization. For the surface fixed heparin, toluidine blue method and APTT method were also used to determine the content and release of heparin in vitro. At the same time, the core-shell fiber surface is heparin-modified PCL, which has been investigated for cytocompatibility and toxicity in our previous work (Hui X, et al. J Biomater Appl. 2020; 34 (6) : 812-826; Xiao Y, et al. Biomater Adv. 2024). In summary, as a small-caliber blood vessel, the characterization of the above properties can basically conclude that the core-shell fiber vascular graft in this submission can be used for carotid artery vascular transplantation in New Zealand white rabbits.

3. In vivo experiments excluded the control experiment. If stable patency rates and tissue regeneration can be induced by blood vessels made of Core-shell fiber, this should be discussed in comparison with the results of other types of fibers, such as those with no core-shell fibers as a control.

Response: Thank you for your professional comments. We did conduct a long-term experiment of core-shell structure fiber and pure PCL fiber vascular graft in a rat model of abdominal aortic vascular transplantation and reported the corresponding experimental results (Xiao Y, et al. Biomater Adv. 2024; 165-214018; Fang Z., et al., Biomater.Sci.10 (2022) 4257-4270), small diameter vascular grafts using core-shell structure fibers have shown higher vascular patency rates and better elastin remodeling in the same animal model. Therefore, on the basis of this experiment, the carotid artery of New Zealand white rabbits was transplanted. The results of the experiment were compared with those of the double-layer vascular grafts performed earlier by our research group (Jin X, et al. Macromol Biosci. 2019; 19(8):e1900114; Hui X, et al. J Biomater Appl. 2020; 34(6):812-826) and surface heparinized vascular grafts have shown better resistance to vascular aneurysm occurrence, elastin remodeling, and vascular patency. At the same time, due to animal ethical considerations, pure PCL fibrovascular was not set as a control.

Reviewer #2:

1.Manuscript dosent have page and line no.

Response: We have added page and line number in the revised version.

2. In section 2.1 Materials - sentence "N-Hydroxy succinimide was purchased from Sigma (St. Louis, MO, USA) was purchased from was purchased from Innochem (Beijing, China)" looks incomplete.

Response: We have completed this sentence.

3. In the same section please provide Ethics approval no.

Response: The Ethics approval no. have been provided in the part of “Surgical procedures”.

4. In section 2.2 Fabrication please clearly claim which polymer is forming core and which is forming shell

Response: We have modified the part 2.2 in order to explain the core-shell structure clearly.

5. In section 2.3 authors used many short forms (EDC, DIPEA, HOBT, NHS, MES etc); it might be from their previous publication, please provide full form of the same in this manuscript

Response: We have provided full form of these short forms in the revised version.

6.In section 2.4 and 2.5 authors should include more characterization like DCS, XRD etc to improve understanding on thermal and crystalline behavior of the scaffolds

Response: Thanks for your comment. In this paper, we also characterized the surface morphology and core-shell structure of core-shell fibers by SEM and TEM, and studied the mechanical properties and degradation properties of the vascular graft after surface heparinization. For the surface fixed heparin, toluidine blue method and APTT method were also used to determine the content and release of heparin in vitro. At the same time, the core-shell fiber surface is heparin-modified PCL, which has been investigated for cytocompatibility and toxicity in our previous work (Hui X, et al. J Biomater Appl. 2020; 34 (6) : 812-826; Xiao Y, et al. Biomater Adv. 2024). In summary, as a small-caliber blood vessel, the characterization of the above properties can basically conclude that the core-shell fiber vascular graft in this submission can be used for carotid artery vascular transplantation in New Zealand white rabbits.

7. In section 2.6 Why author used type I Collagenase for degradation of polymers?

Response: Thanks a lot for this suggestion. This experiment was initially designed to investigate whether vascular grafts could be affected by matrix metalloproteinases during regeneration in vivo, and accordingly the more common type I collagenase was chosen as the active component of enzymatic degradation in vitro.

8. in the same section "wt is the mass before degradation after accelerated degradation for 7 days" looks incomplete

Response: We have revised this sentence.

9.Section 2.8 Please check "werre was..")

Response: We have revised this sentence.

10. Section 2.9.1 authors mentioned that they kept SDVG in 75% alcohol for 20 min. have they studied the effect of alcohol on HCS? please compare it using FTIR

Response: We are very sorry that due to the existing experimental conditions and the limitation of raw materials, we cannot re-prepare the material and carry out this test at present. But based on our previous experience, alcohol immersion has no effect on the material structure.

11. Figure 1 Please provide the cross section of core-shell fiber

Response: We are very sorry that due to the existing experimental conditions and the limitation of raw materials, we cannot re-prepare the material and recapture the picture of the materials.

12. Figure 4 (C-D) it should be (C-F)

Response: We have revised this sentence.

13. Just before figure 6 authors mentioned (Figure 8); there is no figure 8 in the manuscript

Response: We have revised it in the paragraph.

14. In discussion please provide full form of VSMC

Response: we have provided the full name of VSMCs in the paragraph. Vascular smooth muscle cells (VSMCs).

---

## [Decision Letter · Decision Letter 1]

21 Nov 2024

Dear Dr. wang,

These two comments need special attention:

Reviewer #1: In general, results from animal experiments without control experiments are unacceptable in a scientific paper. The results using the grafts made with pure PCL in rabbits should be compared. Previous data using other animal species cannot be used as controls.

Reviewer #2: The previous publication information should be properly linked to the current manuscript to strengthen the claim of continuation of the study.

We look forward to receiving your revised manuscript.

Kind regards,

Pradeep Kumar, Ph.D.

Academic Editor

PLOS ONE

Journal Requirements:

Reviewers' comments:

Reviewer's Responses to Questions

**Comments to the Author**

Reviewer #1: (No Response)

Reviewer #2: All comments have been addressed

2. Is the manuscript technically sound, and do the data support the conclusions?

Reviewer #1: (No Response)

Reviewer #2: Yes

3. Has the statistical analysis been performed appropriately and rigorously?

Reviewer #1: (No Response)

Reviewer #2: Yes

4. Have the authors made all data underlying the findings in their manuscript fully available?

Reviewer #1: (No Response)

Reviewer #2: No

5. Is the manuscript presented in an intelligible fashion and written in standard English?

Reviewer #1: (No Response)

Reviewer #2: Yes

Reviewer #1: In general, results from animal experiments without control experiments are unacceptable in a scientific paper. The results using the grafts made with pure PCL in rabbits should be compared. Previous data using other animal species cannot be used as controls.

Reviewer #2: The previous publication information should be properly linked to the current manuscript to strengthen the claim of continuation of the study

**Do you want your identity to be public for this peer review?** For information about this choice, including consent withdrawal, please see our Privacy Policy

Reviewer #1: No

Reviewer #2: **Yes: ** Ravindra Badhe

---

## [Author Response · Author response to Decision Letter 2]

5 Mar 2025

Reviewer #1: In general, results from animal experiments without control experiments are unacceptable in a scientific paper. The results using the grafts made with pure PCL in rabbits should be compared. Previous data using other animal species cannot be used as controls.

Response Thanks for your conductive suggestions. We reapplied 4 more rabbits to verify pure PCL grafts in rabbit carotid artery grafting model. And the results of patency and livability were shown in Figure S1. Moreover, the regeneration of new tissue was characterized by tissue staining and immunohistochemistry.

Reviewer #2: The previous publication information should be properly linked to the current manuscript to strengthen the claim of continuation of the study.

Response Thanks for your conductive suggestions. We have linked our previous publication to this paper in the part of “Discussion.”

---

## [Decision Letter · Decision Letter 2]

24 Mar 2025

Dear Dr. wang,

We look forward to receiving your revised manuscript.

Kind regards,

Pradeep Kumar, Ph.D.

Academic Editor

PLOS ONE

Journal Requirements:

Additional Editor Comments:

The authors responded that "We reapplied 4 more rabbits to verify pure PCL grafts in rabbit carotid artery grafting model. And the results of patency and livability were shown in Figure S1." The authors should provide information on revised ethical approval details as well as incorporate Figure S1 in the main manuscript and discuss the same with the test group.

Reviewers' comments:

Reviewer's Responses to Questions

**Comments to the Author**

Reviewer #2: All comments have been addressed

2. Is the manuscript technically sound, and do the data support the conclusions?

Reviewer #2: Yes

3. Has the statistical analysis been performed appropriately and rigorously?

Reviewer #2: Yes

4. Have the authors made all data underlying the findings in their manuscript fully available?

Reviewer #2: Yes

5. Is the manuscript presented in an intelligible fashion and written in standard English?

Reviewer #2: Yes

Reviewer #2: My queries are answered thank you

---

## [Author Response · Author response to Decision Letter 3]

2 Sep 2025

Reviewer #1: In general, results from animal experiments without control experiments are unacceptable in a scientific paper. The results using the grafts made with pure PCL in rabbits should be compared. Previous data using other animal species cannot be used as controls.

Response Thanks for your conductive suggestions. We reapplied 4 more rabbits to verify pure PCL grafts in rabbit carotid artery grafting model. And the results of patency and livability were shown in Figure S1. Moreover, the regeneration of new tissue was characterized by tissue staining and immunohistochemistry.

Reviewer #2: The previous publication information should be properly linked to the current manuscript to strengthen the claim of continuation of the study.

Response Thanks for your conductive suggestions. We have linked our previous publication to this paper in the part of “Discussion.”

---

## [Editor Report · Decision Letter 3]

12 Oct 2025

Dear Dr. wang,

Thank you for submitting your manuscript to PLOS ONE. After careful consideration, we feel that it has merit but does not fully meet PLOS ONE’s publication criteria as it currently stands. Therefore, we invite you to submit a revised version of the manuscript that addresses the points raised during the review process.

We look forward to receiving your revised manuscript.

Kind regards,

Pradeep Kumar, Ph.D.

Academic Editor

PLOS ONE

Journal Requirements:

1.I f the reviewer comments include a recommendation to cite specific previously published works, please review and evaluate these publications to determine whether they are relevant and should be cited. There is no requirement to cite these works unless the editor has indicated otherwise. 

Additional Editor Comments:

The supplementary Figure S1 is not provided with the revised manuscript and the reapplication ethical clearance certificate should have been provided.

---

## [Author Response · Author response to Decision Letter 4]

30 Oct 2025

1.I f the reviewer comments include a recommendation to cite specific previously published works, please review and evaluate these publications to determine whether they are relevant and should be cited. There is no requirement to cite these works unless the editor has indicated otherwise.

Response Thanks for your conductive suggestions. This article cites the literature previously published by my research team, mainly References 3, 9, 20, and 21. Among them, the vascular preparation methods involved in references 20 and 21 have great reference value for this paper. In this study, similar methods to those in previous studies were applied in the material preparation process, so References 20 and 21 have not been modified. However, after evaluation, we believed that references 3 and 9 could be deleted. We removed them and replaced them with other references.

Response Thanks for your conductive suggestions. The citation format of reference 7 is incomplete. We have made corresponding modifications. Reference 10 was previously a quoted web address. However, a latest reference has now been published, so this new reference has replaced the previous one.

3.The supplementary Figure S1 is not provided with the revised manuscript and the reapplication ethical clearance certificate should have been provided.

Response We have supplemented Figure S1 in the article. Currently, it is numbered Figure 5 in the article. In addition, we have uploaded the new ethical license certificate.

4.Other modifications

This work was supported by the National Key R&D Program of China (2022YFC2409802). We have added information on financial support in the article

In the previous version, the affiliated institutions of some authors were filled in incorrectly. We have made corrections

---

## [Editor Report · Decision Letter 4]

5 Nov 2025

Preparation and in vivo effectiveness evaluation of heparin-loaded PLGA@PCL core-shell fiber small-diameter vascular grafts

PONE-D-24-24489R4

Dear Dr. wang,

We’re pleased to inform you that your manuscript has been judged scientifically suitable for publication and will be formally accepted for publication once it meets all outstanding technical requirements.

Kind regards,

Pradeep Kumar, Ph.D.

Academic Editor

PLOS ONE
---

## [Editor Report · Acceptance letter]

PONE-D-24-24489R4

PLOS One

Dear Dr. wang,

I'm pleased to inform you that your manuscript has been deemed suitable for publication in PLOS One. Congratulations! Your manuscript is now being handed over to our production team.

Kind regards,

on behalf of

Prof. Pradeep Kumar

Academic Editor

PLOS One